# Family Household Characteristics and Stunting: An Update Scoping Review

**DOI:** 10.3390/nu15010233

**Published:** 2023-01-02

**Authors:** Desy Indra Yani, Laili Rahayuwati, Citra Windani Mambang Sari, Maria Komariah, Sherllina Rizqi Fauziah

**Affiliations:** 1Department of Community Health Nursing, Faculty of Nursing, Universitas Padjadjaran, Bandung 45363, Indonesia; 2Department of Fundamental Nursing, Faculty of Nursing, Universitas Padjadjaran, Bandung 45363, Indonesia; 3Study Program of Bachelor of Nursing, Faculty of Nursing, Universitas Padjadjaran, Bandung 45363, Indonesia

**Keywords:** family, household, stunting

## Abstract

Stunting remains a public health concern in developing countries. Factors related to stunting have been categorized using various frameworks. Family plays an important role in providing nutrients for children; however, no review specifies this aspect for identifying family characteristics related to stunting. This study aimed to identify family household characteristics related to stunting among children aged less than 5 years. A scoping review was undertaken with sources from PubMed, CINAHL, and Scopus, using the keywords “family characteristics” AND “growth”. Inclusion criteria were (1) correlational study; (2) published between 2018 and 31 July 2022; (3) families with children under the age of 5 years; and (4) independent variable any measure of stunting factors from family and household factors. Of 376 articles, only 20 met the inclusion criteria of the study. The family household characteristics included individual factors (sex age, history of diarrhea, and birthplace), family factors (family headship, primary caregiver/mother, social-cultural orientation, and family system factors), and environmental factors. Various child variables, family factors, and environmental factors (the type of home, floor type, water access, source of drinking water, and household electricity) were identified as being associated with stunting. Therefore, these factors should be evaluated to prevent and control stunting, and they should be incorporated into health programs targeting stunting.

## 1. Introduction

Malnutrition may be related to poor diet or severe and recurrent infections, especially in disadvantaged populations. Of the three forms of malnutrition (stunting, wasting, and overweight), stunting is the most common nutritional problem experienced by toddlers worldwide. Globally, approximately 149 million children experience stunting. In 2018, more than half of the world’s stunted toddlers came from the Asian continent (81.7 million cases), while more than a third lived in Africa (58.8 million cases) [1]. Based on data from the World Health Organization (WHO), Indonesia has the third highest prevalence of stunting in the Southeast Asia region, with an average prevalence of 36.4% for 2005–2017 [2]. Although the prevalence of stunting has decreased, progress to reduce stunting has not been equal across regions and sub-regions [1].

Stunting is defined by the WHO as a condition where children experience growth retardation due to poor diet or repeated infections at significant risk of experiencing illness or death [3]. Stunting is also defined as a condition of a child with less than expected length or height for their age, being less than −2 standard deviations based on WHO child growth standards. Some toddlers who experience stunting may experience difficulties in achieving optimal physical and cognitive development. In addition, the damage from stunting can last a lifetime and affect the next generation [4].

Various risk factors for stunting have been identified and reviewed. Direct and indirect factors can influence the nutritional status of a child. Direct factors include food, infectious diseases, and child characteristics (male sex, low birth weight (LBW), and food consumption). Meanwhile, indirect factors include non-exclusive breastfeeding, health services, and family characteristics (parents’ occupation, parents’ education, and family’s economic status) [5]. Low parental education levels, particularly mother education, were closely connected with childhood stunting. Although not universally, the likelihood of a child being stunted increased as parental education decreased, and the possibility of stunting was around twice as high for children of parents with the lowest education level compared to those with the greatest [6]. Again, education was one of the most influential factors [7]. Parents who offer the right, appropriate, and frequent nutrition can reduce child undernutrition [8]. In addition, knowledge can influence practical skills and change mothers’ actions in an ideal setting. The WHO has developed a conceptual framework on childhood stunting causes, including household and family factors, inadequate complementary feeding, breastfeeding, and infection [9]. A review used the WHO’s framework and only found poor nutrition during pregnancy, inadequate sanitation and water supply, food insecurity, low caregiver education, household wealth, paternal smoking, mother’s age, and parenting as household and family factors in Indonesia [10]. Another study divided risk factors for stunting into parental factors, children factors, and environmental factors [11], but did not include family system factors. However, no study has examined family characteristics and their influence on stunting.

Family is one factor that influences the growth pattern of children and toddlers in a family. Studies of stunting risk factors with family characteristics usually assess maternal, child, and environmental factors separately [12,13]. Most children who suffer from stunting come from families with low purchasing power, poor housing conditions, no clean water supply that meets health requirements, poor parental education, and unfavorable attitudes and habits [14]. Family household characteristics are part of the social determinant of health as families’ health is essential to the health of family members, communities, and societies. The social determinants of health are non-medical elements that affect health outcomes. They are the conditions under which people are born, grow, work, live, and age and the more extensive set of factors and institutions that shape these settings [15]. Culture, social norms, social policies, and political institutions affect family health and are affected by families. Families integrate and understand sociocultural and political surroundings to socialize and protect their members. Concerns about individuals’ natural worlds can affect family health by protecting or endangering them. Families can reduce the risk for their members, but fragile families are sensitive to changing social and political systems [16]. Therefore, identifying family household characteristics related to stunting is essential to prepare a family-based stunting program.

Family resilience is the ability of a family to use its resources to achieve family independence and prosperity. It is a dynamic condition of a family that has tenacity and toughness and contains physical, material, and psychological-mental-spiritual abilities to live independently, support its members, and achieve a state of harmony in improving physical and spiritual well-being [17]. Thus, family resilience can make the family more prepared and robust in solving the problems and difficulties it faces. Family participation and functioning in increasing family resilience and food security are needed to fulfill the nutritional needs of children under 5 years of age, and the existence of family support in providing nutritious food and the family’s socioeconomic level dramatically affect the nutritional quality of children and children’s health problems, which are therefore dependent on the role of parents and every family member in the family. Based on the explanation above, this study aimed to identify the association of family household characteristics with the incidence of stunting in children.

## 2. Materials and Methods

This scoping review adhered to the framework described by Arskey and O’Malley [18]. The study was conducted in a manner that is methodical, extremely rigorous, and transparent. No review protocol or registration was performed for this scoping review.

The searching process used three databases, PubMed, Cumulative Index to Nursing and Allied Health Literature (CINAHL), and Scopus, and the following keywords: “family characteristics” and “growth disorder” or “stunting” with their synonyms. In addition, articles were searched using tags of the keywords in Mendeley. The search was conducted in August 2022. All studies were uploaded to the reference management platform Mendeley, and identical references were removed. We performed the process at least twice on each of the databases and checked and compared the results to ensure that no pertinent articles were overlooked.

The screening selection of articles was based on key terms and subject headings. The inclusion criteria were (1) classified as a correlation, cohort, or case control study; (2) published between 1 January 2018 and 31 July 2022 in the English or Indonesian language; (3) dealt with families with children aged less than 5 years; and (4) independent variable any measure of stunting factors from family and household factors. Using the inclusion criteria, we screened all the selected literature for titles and abstracts in Mendeley, and this technique was repeated twice. Any identified inconsistency was rectified by closer examination, and each of the selected articles was read in their entirety. Three members of the team reviewed this procedure.

## 3. Results

Based on the search performance results in three databases, 376 articles were acquired and imported into Mendeley. A total of 66 duplicates were removed, resulting in 310 articles in the selection process. A total of 20 articles were selected on the basis of title and abstract suitability (Figure 1).

### 3.1. Study Characteristics

Publication numbers were greatest for studies conducted in Indonesia (55%), in 2019 (40%), or with a cross-sectional study approach (45%). The various sampling techniques were represented by approximately the same number of articles. However, the sample size varied from 40 to 384,928 children. The publications involved children under 5 years of age, with the majority being toddlers (85%) (Table 1).

### 3.2. Stunting Prevalence

The lowest prevalence of stunting was 4.8%, but that study involved stunting in overweight children, and one study did not mention the stunting prevalence. The prevalence of stunting in the remaining studies ranged from 20% to 71%. All studies used the WHO’s standard to measure stunting, as well as secondary data and direct measurements (Table 2).

### 3.3. The Correlation between Family Household Characteristics and The Prevalence of Stunting

The family household characteristics included individual factors, family factors, and environmental factors. Individual factors consisted of child characteristics: gender, age, history of diarrhea, and birthplace. Gender of the child (75%), age of the child (40%), and history of diarrhea (100%) contributed to stunting. The birthplace of the child did not affect stunting (Table 3).

Family factors comprised family headship, primary caregiver/mother, social-cultural orientation, and family system factors. Three out of four articles showed gender and education of the household headship correlated with stunting. However, the ages of the household headship and primary caregiver were not associated with stunting. Three of five articles found that extended family contributed to stunting. The gender of the primary caregiver, the mother’s height, and the mother’s marital status did not contribute to stunting, while the mother’s age (25%), mother’s knowledge (100%), education of the primary caregiver (55%), and mother’s occupation (40%) contributed to stunting. The relationship between the primary caregiver and the child was examined in two articles and was found to be not correlated or significantly correlated. Of 16 articles that studied family wealth, 62.5% showed a link with stunting. Two studies considered family ethnicity and found either a significant or an insignificant relationship between family ethnicity and stunting. Four articles that included the area of residence showed a significant connection Only one in five articles showed that the number of family members affected stunting. The number of dependent adults did not contribute to stunting, while the number of children (100%) and siblings aged under 5 years contributed to stunting. Family awareness and family wellness did not relate to stunting. Food consumption (100%); food security (50%); knowledge, attitude, and behavior toward nutrition (100%); healthy parenting (100%); and family quality of life and its domain contributed to stunting (Table 3).

Of the environmental factors, the type of home, floor type, water access, source of drinking water, and household electricity contributed significantly to stunting. However, one in three articles showed no correlation between sanitation facilities and stunting (Table 3). A summary of the essential variables from family household characteristics is explained in Figure 2.

## 4. Discussion

This scoping review categorized family household characteristics that contribute to stunting into individual, family, and environmental factors. The WHO has developed a conceptual framework on childhood stunting causes, including household and family factors, inadequate complementary feeding, breastfeeding, and infection [36]. This framework classifies maternal factors and the home environment as part of the household and family factors. Of the factors reviewed in the current study, only sanitation and water supply, food insecurity, and caregiver education are included in the home environment category. A previous review used the WHO’s framework and only found poor nutrition during pregnancy, inadequate sanitation and water supply, food insecurity, low caregiver education, household wealth, paternal smoking, mother’s age, and parenting as household and family factors in Indonesia In addition, risk factors for stunting were divided into parental factors, children factors, and environmental factors [37] but that study did not include family system factors. However, family and household characteristics need to be determined, together with whether they contribute to stunting.

For child characteristics, gender and history of diarrhea were consistent risk factors for stunting. A prior review also revealed that children’s gender and infection contributed to stunting [37]. Although stunting was shown to affect 52.3% of young females and 47.7% of young men [38], other studies revealed that boys had a much higher risk of stunting than girls [39]. A diarrheal episode was shown to be a risk factor for stunting [40,41].

The age of the child showed divergent findings in the present study. This differs from a past review that found that children’s generation was related to stunting [37]. A study showed that stunting was most prevalent in children between the ages of 25 and 36 months, at 51.7% [42] but other studies found infants aged 12–23 months had a higher prevalence rate (40.4%) [43,44]. Another study found that stunting was most common in the age group of 49–59 months, where it was 65.5%, and least common in the age group of 6–12 months [38]. The other study found that increasing child’s age was a risk factor for stunting [40].

Household headship characteristics of gender and education were essential factors for stunting. This result differs from a prior study that showed that even after adjusting for family wealth, education, and residence, there were no changes in stunting prevalence based on household headship [45]. Most female-headed households regard themselves as vulnerable and less stable livelihoods [46] because of their often lower earnings and restricted access to vital services [45]. Multiple factors, including divorce, widowhood, labor mobility of husbands, polygyny, matriarchal social structures, and non-marital childbirth, may have diverse effects on household dynamics and resources when a woman heads a home. This variation among female-headed households may result in variable child health, nutrition outcomes, and intervention coverage. However, the most important lesson is not that mothers are irresponsible parents but that females in most communities worldwide have restricted access to resources.

Other family factors, extended family, area of residence, family wealth, number of children, and number of siblings aged under 5 years were associated with stunting. Family economic status was a risk factor for stunting in developing countries [6,37,40]. However, family ethnicity did not affect stunting. In addition, living in rural areas was a risk factor for stunting [6,40]. Children who lived in homes with three or more children aged under 5 years and families with five to seven people considerably increased their risk of stunting [44].

The mother’s knowledge and education were primary factors for mother characteristics. The mother’s education was a risk factor for stunting [37,41], and poor maternal education was associated with stunting in Indonesia [6]. Therefore, research suggests enhancing mother understanding regarding exclusive breast milk and supplemental feeding, comprehensive immunization, and contagious disease avoidance [47,48] to ensure improvement of affective and psychomotor aspects [49]. Further, superior parental educational attainment was consistently found to be a strong predictor of improved child growth outcomes [4,40]. Other irrelevant factors were the gender of the primary caregiver, mother’s height, marital status, mother’s age, and relationship between the primary caregiver and the child. However, some reviews found differently. For example, some studies revealed that the mother’s height [6,37] and age [40] were risk factors for stunting.

Food consumption; food security; knowledge, attitude, and behavior toward nutrition; healthy parenting; and family quality of life and its domain contributed to stunting. Family awareness and family wellness did not relate to stunting. Food security is related to the socioeconomic factors of the family, and a past review found that food security and lack of access to suitable nutrition contribute to stunting, with strong evidence in Bangladesh [41].

Health environment factors, including the type of home, floor type, water access, source of drinking water, and household electricity are associated significantly with stunting. Interestingly, a previous review had adequate supporting data linked to childhood stunting, including lack of sanitation, improper community waste disposal, dirty flooring in homes, mycotoxins in food, and indoor solid fuel burning [50]. Two studies examined the availability of clean water sources, and their findings produced robust conclusions in the present study. Untreated drinking water was associated with stunting in Indonesia [6,40]. Conversely, a previous systematic review had conflicting findings, and the conclusions are still up for debate [50]. The importance of the sanitation facility remained inconclusive in the present study because of the inconsistent results. However, inadequate sanitation was a risk factor for stunting among children in developing countries [6,37]. Other environmental factors related to stunting included ambient PM2.5 particle levels and household air pollution exposure [51]. Only one to three articles studied each variable of environmental factors in the present review, as most articles did not include environmental factors as part of family characteristics. Environmental factors such as lack of sanitation, dirt floor, and waste disposal are not direct causes of stunting. However, these factors increase the risk of infection that results from undernutrition, which may then lead to stunted growth in children. Encouraging people and participation were the most successful methods for empowering individuals to promote healthy and clean behavior [52].

Stunting is a multidimensional factor. There are no specific categories for each element as this is affected by the underpinning theory, and there is variability in the significance and magnitude of the relationships. Some factors are modifiable, and a lack of information with regard to the effects of altering these factors presents research gaps, which warrant subsequent analyses.

Because not all families can sustain stunting adjustments, their vulnerability grows. Family resilience is appropriate and persistent access to money and resources to meet basic requirements such as food, clean water, health services, education, housing, time to engage in society, and social integration [53]. Family household characteristics are essential factors of social determinants of health. However, not all individuals, families, communities, or populations experience inferior health outcomes despite being exposed to negative socioeconomic determinants of health [54]. Protective social elements such as social support are predicted to prevent or moderate negative consequences, whereas resilience mechanisms provide adaptive functioning in response to adverse exposures [55].

Family resilience is crucial since it’s the foundation for extraordinary human resources. Resilient families can also fulfill societal duties. Family resilience reduces social and economic problems caused by dysfunction or disintegration. The family can be considered one of the nurse’s clients. Family plays a vital role in caring for every family member, including the children susceptible to stunting. As stunting is a preventable nutrition problem, identifying family factors and considering them in prevention will favor children’s development. Developing a partnership with religious and community leaders enhances prevention and care [56].

### 4.1. Limitations

There are some limitations to our scoping review. In this study, only three databases and the Mendeley were used as access to find the sources of the articles being reviewed. Articles, especially from Asian continent backgrounds, were dominated by reports from Indonesia. So, there may still be articles that can be obtained from another country, especially in countries with a high prevalence of stunting, to better describe the conditions associated with stunting. We also put only three keywords and their synonyms when searching the articles. Meanwhile, other keywords should also be added to maximize the article’s search process.

### 4.2. Implications for Clinical Practice

The results of this scoping review show various factors that can contribute to and associate with stunting. Knowing the individual, family, and environmental factors, as the family household characteristics related to stunting, can be a source of reference to carry out integrated prevention between the community and health workers, especially nurses. Thus, prevention programs for stunting can be carried out by reaching more possibilities.

## 5. Conclusions

Several family home features as part of the social determinants of health, including individual factors, family factors, and environmental factors, were identified as related to stunting in this scoping review. Hence, the social determinants of health need to be considered to improve stunting rates.

## Figures and Tables

**Figure 1 nutrients-15-00233-f001:**
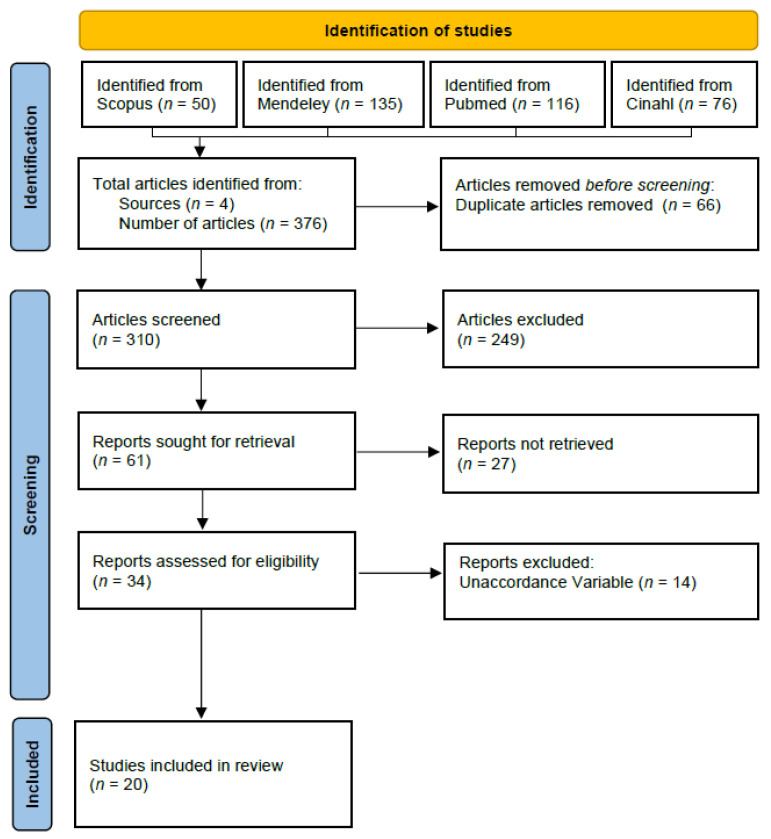
Prisma Flow Diagram.

**Figure 2 nutrients-15-00233-f002:**
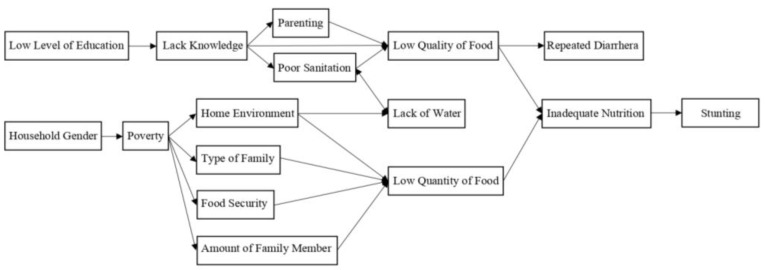
Family Household Characteristics and Stunting.

**Table 1 nutrients-15-00233-t001:** Study Characteristics (*n* = 20).

Components	Author	Frequency
Publishing Year		
2017	[19,20,21]	3
2018	[22]	1
2019	[12,23,24,25,26,27,28,29]	8
2020	[30,31]	2
2021	[9,11,32,33,34]	5
2022	[35]	1
Country		
Ethiopia	[11]	1
Indonesia	[9,22,24,25,26,28,29,32,33,34,35]	11
Kwaluseni	[31]	1
Mozambique	[20]	1
Pakistan	[19]	1
Rwanda	[12,23]	2
Sub-saharan Africa	[27]	1
Uganda	[21,30]	2
Study Method		
Case control	[20,21,22,32,34,35]	6
Cohort Study	[11]	1
Cross Sectional Study	[9,12,19,23,24,25,26,28,29,30,33]	9
Survey	[27,31]	2
Sampling Technique		
Random Sampling		
Cluster Sampling	[19,26,27,28]	4
Random sampling	[12,21,29]	3
Systematic Sampling	[30,31]	2
Two stage sampling	[11,23]	2
Non Random Sampling		
Accidental Sampling	[24]	1
Consecutive Sampling	[20,33]	2
Purposive Sampling	[9,22,32,34,35]	5
Total Population Sampling	[25]	1
Sample Size		
<50	[34]	1
<200	[21,22,25,26,28,32,35]	7
<1000	[11,20,29,30,33]	5
<10,000	[12,19,23,24,31]	5
>10,000	[9,27]	2
Respondent’s Age (Children)		
Infants		
0–6 months	[12,20,22,27,33]	5
6–12 months	[11,12,20,21,22,26,27,30,31,33]	10
Toddler (1–3 years)	[9,11,12,19,20,21,22,26,27,29,30,31,32,33,34,35]	17
Preschool (3–5 years)	[9,11,12,19,20,21,24,26,27,30,31,32,33,34,35]	15
Under five years old	[23,28,29]	3
School (above 5 years)	[25]	1

**Table 2 nutrients-15-00233-t002:** Stunting Prevalence (*n* = 20).

Components	Author	Frequency
Stunting Prevalence (n (%))		
≤25%		
949 (4.8%) (stunting & overweight)	[9]	1
512–593 (20%–24%) (different years (2010 and 2014)	[31]	1
31 (25%)	[25]	1
26%–50%		
32 (31.37%)	[24]	1
36 (32.4%)	[29]	1
54 (32.73%)	[26]	1
56 (33%)	[21]	1
73.751 (34%)	[27]	1
102 (36%)	[20]	1
74 (38.5%)	[28]	1
1.355 (37%)	[12]	1
1.408 (43%)	[11]	1
167 (44.9%)	[30]	1
198 (46.7%)	[33]	1
57 (50%)	[35]	1
60 (50%)	[32]	1
20 (50%)	[34]	1
>51%		
94 (51%)	[22]	1
5.140 (71%)	[19]	1
Not Mentioned	[23]	1
Stunting Assessment Method		
KMS Data (Indonesian Local Card)	[22,24,32,35]	4
Demographic and Health Survey (DHS)	[12,27]	2
Height/ Length Assessment	[23,25,26,30]	4
Height for age score	[9,11,19,20,21,29,31,33]	8
Not Mentioned	[28,34]	2

**Table 3 nutrients-15-00233-t003:** The Correlation between Family Household Characteristics and Stunting Prevalence (*n* = 20).

Family Household Variables	Author	Statistic Test	*p*-Value	Correlation Score	Correlation
**Individual Level Factors**					
Sex of child	[20]	Binary Logistic Regression (Crude Odds Ratio)	0.001	4.01	Significantly correlated
	[21]	Chi-square	0.016	n/a	Significantly correlated
	[12]	Chi-square	<0.01	n/a	Significantly correlated
	[27]	Chi-square	<0.001	n/a	Significantly correlated
	[31]	Chi-square	<0.001	15.61	Significantly correlated
	[9]	Chi-square	>0.05	0.89	Not correlated
	[11]	Logistic linear	>0.05	1.52	Not correlated
History of diarrhoea	[12]	Chi-square	<0.01	n/a	Significantly correlated
Age of child	[19]	Logistic regression (Odds Ratio)	0.96	1	Not correlated
	[21]	Chi-square	0.798	n/a	Not Correlated
	[22]	n/a	0.640	n/a	Not Correlated
	[23]	Chi-square	0.0001	1.78	Significantly correlated
	[12]	Chi-square	<0.01	n/a	Significantly correlated
	[27]	Chi-square	<0.001	n/a	Significantly correlated
	[33]	Product Moment Test	>0.05	0.027	Not correlated
	[28]	Chi-square	3.92	1.348	Not correlated
	[31]	Chi-square	<0.001	51.26	Significantly correlated
	[11]	Logistic linear	>0.05	1.43	Not correlated
Birth place of child	[21]	Chi-square	1.0	n/a	Not correlated
**Family Level Factors**					
Household headship					
Gender of house hold headship	[19]	Logistic regression (Odds Ratio)	<0.01	0.74	Significantly correlated
	[21]	Chi-square	1.00	n/a	Not correlated
	[27]	Chi-square	<0.001	n/a	Significantly correlated
	[9]	Chi-square	>0.05	0.83	Not correlated
Education of household Headship	[19]	Logistic regression (Odds Ratio)	0.01	0.56	Significantly correlated
	[22]	Chi-square	0.003	n/a	Significantly correlated
	[28]	Chi-square	0.000	4.596	Significantly correlated
	[9]	Chi-square	>0.05	0.68	Not correlated
Age of household headship	[28]	Pearson	0.132	0.069	Not correlated
	[9]	Chi-square	>0.05	0.82	Not correlated
Primary Caregiver					
Age of parent	[33]	Product Moment Test	>0.05	0.049	Not correlated
Type of Family	[20]	Binary Logistic Regression (Crude Odds Ratio)	18.36	0.001	Significantly correlated
	[22]	Chi-square	0.017	n/a	Significantly correlated
	[27]	Chi-square	<0.001	n/a	Significantly correlated
	[28]	Chi-square	0.059	1.841	Not correlated
	[9]	Chi-square	>0.05	0.98	Not correlated
Sex of primary caregiver	[21]	Chi-square	0.378	n/a	Not correlated
Education of primary caregiver	[21]	Chi-square	0.126	n/a	Not correlated
	[23]	Chi-square	0.937	1.42	Not correlated
	[12]	Chi-square	<0.01	n/a	Significantly correlated
	[24]	Chi-square	0.455	n/a	Not correlated
	[25]	Chi-square	0.00	n/a	Significantly correlated
	[27]	Chi-square	<0.001	n/a	Significantly correlated
	[28]	Chi-square	0.006	2.440	Significantly correlated
	[29]	Chi-square	0.048	n/a	Significantly correlated
	[31]	Chi-square	<0.001	59.53	Significantly correlated
	[34]	Chi-square	0.000	22.667	Significantly correlated
	[11]	Logistic linear	>0.05	0.19	Not correlated
Relationship of primary caregiver to child	[21]	Chi-square	0.693	n/a	Not correlated
	[23]	Chi-square	0.01	0.82	Significantly correlated
Mother’s height	[24]	Chi-square	0.257	n/a	Not correlated
	[34]	Chi-square	0.84	1.08	Not correlated
Mother’s knowledge	[24]	Chi-square	0.003	n/a	Significantly correlated
	[34]	Chi-square	0.006	28	Significantly correlated
Mother’s marital stutus	[27]	Chi-square	<0.001	n/a	Not correlated
	[31]	Chi-square	0.195	3.27	Not correlated
Mother’s age	[27]	Chi-square	<0.001	n/a	Significantly correlated
	[28]	Pearson	0.109	0.065	Not correlated
	[31]	Chi-square	0.390	6.30	Not correlated
	[11]	Logistic linear	>0.05	0.97	Not correlated
Mother’s occupation	[20]	Binary Logistic Regression	0.304	0.23	Not correlated
	[28]	Chi-square	0.005	1.045	Significantly correlated
	[29]	Chi-square	0.797	n/a	Not correlated
	[34]	Chi-square	0.744	1.238	Not correlated
	[11]	Logistic linear	<0.001	0.09	Significantly correlated
Socio-Cultural Orientation					
Famliy wealth/socio-economic status	[19]	Logistic regression (Odds Ratio)	<0.01	0.35–0.78	Significantly correlated
	[21]	Chi-square	0.510	n/a	Not correlated
	[22]	Chi-square	0.081	n/a	Not correlated
	[23]	Chi-square	0.088	1.43	Not correlated
	[12]	Chi-square	<0.01	n/a	Significantly correlated
	[24]	Chi-square	0.320	n/a	Not correlated
	[25]	Chi-square	0.00	n/a	Significantly correlated
	[27]	Chi-square	<0.001	n/a	Significantly correlated
	[28]	Chi-square	0.000	3778	Significantly correlated
	[29]	Chi-square	0.000	n/a	Significantly correlated
	[31]	Chi-square	<0.001	48.30	Significantly Correlated
	[32]	Chi-square	0.004	9.33	Significantly Correlated
	[9]	Chi-square	>0.05	0.73	Not correlated
	[34]	Chi-square	0.001	13.22	Significantly correlated
	[11]	Logistic linear	>0.05	4.01	Not correlated
Family Ethnicity	[19]	Logistic regression (Odds Ratio)	0.02	1.35	Significantly correlated
	[21]	Chi-square	0.759	n/a	Not correlated
Area of residence	[20]	Binary Logistic Regression (Crude Odds Ratio)	0.001	137.95	Significantly correlated
	[12]	Chi-square	<0.01	n/a	Significantly correlated
	[27]	Chi-square	<0.001	n/a	Significantly correlated
	[31]	Chi-square	0.012	6.38	Significantly correlated
Number of family members	[22]	*t*-test	0.240	n/a	Not correlated
	[12]	Chi-square	0.73	n/a	Not correlated
	[24]	Chi-square	0.245	n/a	Not correlated
	[26]	Chi-square	0.557	n/a	Not correlated
	[9]	Chi-square	<0.01	0.59	Significantly correlated
Number of children	[31]	Chi-square	<0.001	20.63	Significantly correlated
	[9]	Chi-square	<0.05	0.52	Significantly correlated
Number of dependent adult	[9]	Chi-square	>0.05	0.66	Not correlated
Siblings under 5 years old	[20]	Binary Logistic Regression (Crude Odds Ratio)	0.001	41.75	Significantly correlated
Family System Factors					
Family awareness nutrition	[33]	Product Moment Test	>0.05	−0.048	Not correlated
	[11]	Logistic linear	>0.05	1.07	Not correlated
Food consumption score	[21]	Chi-square	0.040	n/a	Significantly correlated
Food security status	[21]	Chi-square	0.021	n/a	Significantly correlated
	[23]	Chi-square	0.006	1.68	Significantly correlated
	[21]	Chi-square	0.004	0.23	Significantly correlated
	[11]	Logistic linear	>0.05	0.72	Not correlated
Family wellness	[33]	Product Moment Test	>0.05	−0.055	Not correlated
Knowledge of nutrition	[28]	Chi-square	0.001	2.971	Significantly correlated
Atttude toward nutrition	[28]	Chi-square	0.001	2.971	Significantly correlated
Behavior toward nutrition	[28]	Chi-square	0.001	2.890	Significantly correlated
Nutritional parenting	[28]	Chi-square	0.000	3.896	Significantly correlated
Family Quality of Life	[33]	Product Moment Test	<0.01	0.209	Significantly correlated
Family interaction	[33]	Product Moment Test	<0.01	0.203	Significantly correlated
Parenting	[33]	Product Moment Test	<0.01	0.175	Significantly correlated
	[25]	Chi-square	0.007	n/a	Significantly correlated
Emotional welfare	[33]	Product Moment Test	<0.01	0.193	Significantly correlated
Physical welfare	[33]	Product Moment Test	<0.01	0.216	Significantly correlated
Family support	[33]	Product Moment Test	<0.01	0.136	Significantly correlated
	[35]	Chi-square	0.49	0.49	Not correlated
**Environment Factors**					
Type of home: Straw and wood	[20]	Binary Logistic Regression (Crude Odds Ratio)	0.002	3.27	Significantly correlated
Exposure to cigarette smoke	[35]	Chi-square	0.09	0.31	Not correlated
Type of floor: soil	[20]	Binary Logistic Regression (Crude Odds Ratio)	0.001	0.05	Significantly correlated
Cooking fuel: wood	[20]	Binary Logistic Regression (Crude Odds Ratio)	0.001	45.5	Significantly correlated
Water access	[23]	Chi-square	0.040	1.52	Significantly correlated
	[12]	Chi-square	<0.01	n/a	Significantly correlated
Sanitation facility	[12]	Chi-square	0.052	0.40	Not correlated
	[30]	Chi-square	<0.01	n/a	Significantly correlated
	[31]	Chi-square	<0.001	32.79	Significantly correlated
Source of drinking water	[21]	Chi-square	0.004	0.92	Significantly correlated
	[31]	Chi-square	0.020	5.45	Significantly correlated
Household electricity	[31]	Chi-square	<0.001	15.48	Significantly correlated

## Data Availability

Not applicable.

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
