# Peer review of "Family Household Characteristics and Stunting: An Update Scoping Review"

_nutrients, 2023, doi:10.3390/nu15010233_

Round 1

Reviewer 1 Report

Family household characteristics are important to study, and this study has some valuable variables to inform this study however family household characteristics should be considered to be some aspects of the social determinants of health to enhance understanding and to facilitate future literature searching to find this paper. Another reason for describing them as the social determinants of health is that without doing so can blame the families while the social determinants of health and these “family household characteristics” are usually not easily rectified by the families as they are due to larger social factors in the individual’s communities. The World Health Organization should be cited regarding the social determinants of health. As your reference Vaivada states: “Household income is an important measure of a household's capacity to afford important elements related to improved nutrition such as food, water, sanitation, and medical care. Compared with other determinants, improvements in asset index consistently predicted some of the greatest improvements in HAZ across the countries analyzed.”

The reason for “female-headed households may result in variable child health” is not that mothers are negligent parents but rather that females in most communities in the world have limited access to resources.  

The authors tried to look at the bigger picture by referring to direct and indirect factors and resiliency. A reason for the association between the “direct” factors including “non-exclusive breastfeeding” with stunting is that all of these are influenced by the social determinants of health. Resiliency is very important, but it needs to be considered whether those with resiliency have better social determinants of health.

The introduction focuses on education and knowledge as the solution to the problem of stunting “Therefore, research suggests enhancing mother understanding regarding exclusive breast milk and supplemental feeding, comprehensive immunization, and contagious disease avoidance [42,43] with ensure improvement of affective and psychomotor aspects [44].” 

Of concern, the introduction refers to associations noted in observational studies as causal relationships which are not likely causal relationships: “Health environment factors, including the type of home, floor type, water access, source of drinking water, and household electricity, contributed significantly to stunting”. This should be changed to refer  to “associations” or “associated”.

Over the longer term what is needed is research evidence to determine whether interventions can improve stunting rates. The authors should draw directed acyclic graphs (ref: Williams PMID: 29967527) to inform their understanding of relationships, including the social determinants of health. With a directed acyclic graph, the authors are likely to gain a better understanding of how stunting could be improved. 

The authors should recognize that while children with stunted heights are at risk of suboptimal development, using statistical cut-offs (such as <-2 SD) will identify a few children who are small but healthy children, approximately 2%. Therefore it is not appropriate to say that these children “will” “experience difficulties in achieving optimal physical and cognitive development”. It would be more correct to write that “some of these children may”.

It is not accurate to say in the conclusion that “evaluating” these factors can prevent and control stunting or that they can easily “be incorporated into health programs targeting stunting” rather the conclusions could be that these indicators of the social determinants of health are associated with child stunting and therefore the social determinants of health need to be considered in attempts to improve rates of stunting.

Author Response

Dear Reviewer,

We have revised all aspects as suggested.  Thank you for providing meaningful comments and suggestions. 

Reviewer 2 Report

This study used a scoping review approach and identified family household characteristics related to stunting among children aged less than 5 years oldThe authors summarized various child variables, family factors, and environmental factors associated with stunting. This is an interesting topic to shift treatment focus to the family towards stunting. Below are some comments to improve the manuscript.

1.     The Introduction is generally comprehensive and well-written. The authors may review studies conducted in families with children aged less than 5 years old since this is your target group.

2. The authors stated at the end of the Introduction that “this study aimed to identify the association of family resilience and food security with the incidence of stunting in children”. This is not identical to your topic “family characteristics and stunting”. Please revise and justify the study objective.

3. Why did the authors only search with three keywords “family characteristics” and “growth disorder” or “stunting”? what about the other synonyms and antonyms? Could the three words cover all the potential articles?

4. Why did the authors set a criterion of “published between 1 January 2018 and 31 July 2022”? What about those published earlier than 2018?

5. There`s a typo in Figure 1 “Records identified from the database (n=3)”. Please revise it to the correct number.

6. Were all the eligible studies conducted within African and Indonesian districts? What about other Asian and European countries?

7. The authors may need to review related family factors identified in the results section and combine them with those discussed in the Introduction and Discussion sections.

8. Please add a brief discussion of the limitations and implications of the current study.

Author Response

(The authors gave the same response as above.)

Round 2

Reviewer 2 Report

The manuscript has been sufficiently improved to warrant publication.